# Morphometric Measurements Prior to Totally Endoscopic Mitral Valve Repair: Technical and Educational Aspects

**DOI:** 10.3390/jcm14082581

**Published:** 2025-04-09

**Authors:** Marie-Elisabeth Stelzmueller, Daniel Zimpfer, Wilfried Wisser

**Affiliations:** Department of Cardiac and Thoracic Aortic Surgery, Medical University Vienna, 1090 Vienna, Austria; marie-elisabeth.stelzmueller@meduniwien.ac.at (M.-E.S.); daniel.zimpfer@meduniwien.ac.at (D.Z.)

**Keywords:** totally endoscopic, minimally invasive, mitral valve repair, computed tomography

## Abstract

**Objective**: The totally endoscopic approach is on the rise to become the new standard in mitral valve surgery. The aim of this study was to develop a morphometric measurement tool for educational purposes to predict operability with low conversion and high repair rates. **Methods**: From January 2020 to March 2023, 64 patients underwent totally endoscopic mitral valve repair (TE-MVR). Of these, 15 patients were deemed to be unsuitable for TE-MVR due to narrow space and/or anticipated complex repair techniques and underwent repair through sternotomy (MVR-open). Angio-CT scanning was performed for preoperative planning and measurements of the following: the distance between the sternum and the spine (DSS), the distance between the skin incision and the anterior anulus of the mitral valve (DNM) and the intercostal space at the level of the skin incision (ICS). **Results**: The repair rate for all patients was 98.7%. In the TE-MVR group, the conversion rate to sternotomy was 3.1%. The 30-day survival was 100%. The DSS was 130.4 ± 18.8 mm and 108.1 ± 17.3 mm, and the DSM 70.7 ± 12.1 mm and 58.5 ± 13.6 mm in the TE-MVR and MVR-open, respectively (*p* < 0.001). Twenty-one TE-MVR patients were found to be technically demanding due to friction and less freedom to move the instruments. The composite morphometric parameter DSS plus 4xICS minus DNM was 53.3, 39.8 and 25.6 for TE-TMReasy, TE-TMRdemanding and MVR-open, respectively (*p* < 0.05 and *p* < 0.01). **Conclusions**: Surgical skills and a long history of expertise are mandatory to achieve excellent results with a low conversion and high repair rate. The composite morphometric parameter may be an easy tool for educational demands to predict the ease and feasibility of TE-MVR.

## 1. Introduction

Minimally invasive surgery for atrioventricular valve repair, either as an isolated procedure or in combination with atrial septal defect (ASD) closure and MAZE procedures, has become an established approach in cardiac surgery [1,2,3,4]. Over the years, advancements in surgical techniques have led to progressively less invasive procedures, minimizing surgical trauma and enhancing patient recovery. One of the most recent innovations in this field is the adoption of totally endoscopic techniques utilizing high-definition 3D endoscopes [5,6,7]. This approach allows complex valve procedures to be performed through a single working port without rib spreading, significantly reducing postoperative discomfort and improving cosmetic outcomes. Typically, access is gained via a 3 to 4 cm periareolar incision in males or a similar submammary fold incision in females. While 3D endoscopic visualization provides superior anatomical detail compared to direct vision or 2D endoscopic surgery, the reduction in incision size inherently limits instrument maneuverability [8]. This trade-off underscores the importance of precise preoperative planning to ensure optimal surgical outcomes, minimize conversion rates and maintain high repair success rates [9].

A key challenge for less experienced surgeons is the learning curve associated with totally endoscopic mitral valve repair. Preoperative morphometric assessment can play a crucial role in predicting surgical feasibility and guiding procedural strategy. The aim of this study was to develop a morphometric measurement tool for educational purposes, enabling the better preoperative assessment of operability and the anticipated ease of mitral valve repair in a totally endoscopic approach. This tool aims to support surgical training and decision-making, ultimately improving patient outcomes and expanding the adoption of minimally invasive techniques.

## 2. Materials and Methods

From January 2020 to March 2023, all 79 patients who were referred for totally endoscopic mitral valve repair (TE-MVR) were included in this retrospective analysis. The patient demographics are presented in Table 1. All patients routinely underwent a complete clinical workup preoperatively, consisting of ECG, transthoracic echocardiography, left heart catheterization and CT scanning.

A preoperative morphological evaluation was performed by a CTA scan of the aorta using a 256-row dual-source CT system (Siemens Somatom Flash, Siemens Medical Systems, Erlangen, Germany). CTA was performed in the craniocaudal direction from the mid-skull level to the level of the trochanter minor. An iodinated contrast agent (60–110 mL, depending on body weight) was administered through venous access. A bolus tracking technique was applied, and the scan was started automatically with a delay of 10 s (64-slice system) or 15 s (dual-source system) after a threshold of 150 HU was reached in the ascending aorta. Electrocardiography gating was used in the 64-slice system, whereas an ultrafast high-pitch acquisition technique was applied in the dual-source system. Automated tube current modulation was applied to all patients. Axial images were reconstructed at 1 and 3 mm slice thicknesses. In addition, parasagittal and coronary maximum intensity projection reconstructions at a 3.2 mm slice thickness were provided.

CT scanning of all patients was implemented in our routine preoperative workup to determine the access path for cannulation and potential soft or hard plaques in the ascending aorta, which would change the surgical process.

The following morphological parameter data were measured and collected retrospectively (Figure 1, Figure 2 and Figure 3):

DSS–Distance between the sternum and the spine: On an axial image, at the level of the midportion of the mitral valve.

DSM–Distance between the sternum and the anterior annulus (A2) of the mitral valve: On an axial image, at the level of the midportion of the mitral valve.

DSA–Distance between the sternum and aorta: On an axial image, just above the right atrial appendage.

DNM–Distance between the nipple and the anterior mitral anulus: From an axial image view, the distance between the nipple as the entering point of the thorax and the anterior mitral anulus was measured. In female patients, the expected entering point through the fourth intercostal space with the tissue above was taken.

DVM–Distance to the left between the vertical and the anterior mitral anulus (DVM): From an axial image view, at the level of the midportion of the mitral valve, the distance between the vertical line of the DSS measurement and the anterior mitral anulus was measured. This is a parameter reflecting how far left the mitral valve is located in the thorax.

AMH–Angle between the mitral valve and the horizontal: From an axial image view, the angle between the mitral valve and the horizontal was measured. The plain of the mitral valve was defined by the connection line between the anterior and posterior anulus at the midportion (corresponding to A2, P2 segment) of the valve.

ANM–Angle between the mitral plane and the line from the working port to the mitral valve: From an axial image view, the angle between the mitral plane and the line from the working port to the mitral valve was measured.


ICS–Width of the intercostal space:


In multiplanar reformatting (MPR), the width of the corresponding intercostal space just below the nipple reflecting the entry site was measured.

Besides the morphologic measures on CT scanning, the decision of the appropriate surgical approach was made on clinical grounds and to the discretion of the surgeon’s experience.

Patients underwent totally endoscopic mitral valve surgery according to our institutional protocol. Briefly, under general anesthesia and double-lumen tube intubation, a periaureolar skin incision was carried out in male patients. Depending on the individual size of the nipple, cutting half of the circumference resulted in a 3 to 4.5 cm skin incision. In female patients, a skin incision of equal length was performed in the submammary fold. The breast tissue was mobilized cranially to reach the level of the fourth intercostal space. In both genders, the underlying intercostal muscle in the fourth intercostal space was cut to the same extent accordingly. A soft tissue retractor (Alexis Wound protector size XS; Applied Medical Resources Corp., Rancho Santa Margarita, CA, USA) was introduced and served as the single working port. No other retractor or rib spreader was used. Then, a 10 mm video port was inserted into the fourth intercostal space in the anterior axillary line. After full heparinization, the patient was cannulated through the femoral artery and vein in a fully percutaneous fashion using Seldinger’s technique, placing two Proglide systems (Perclose Proglide System, Abbott Vascular, Redwood City, CA, USA) on the artery. For arterial cannulation, a cannula with 19 French or less was chosen. In the case of a small femoral artery, an additional 8 French cannula was inserted to commence antegrade leg perfusion. Cardiopulmonary bypass was started, the lungs disconnected from ventilation and a 3D Einstein videoendoscope (Aeskulap, B. Braun, Austria) was inserted. After opening the pericardium, an antegrade cardioplegia line was inserted into the ascending aorta and brought through the opening of the single working port. A Chitwood clamp (Cardiomedical GmbH, Langenhagen, Germany) was inserted through a stab incision in the second intercostal space lateral to the midclavicular line, the ascending aorta was cross-clamped and cardioplegic arrest was induced with cold blood cardioplegia, which was readministered every 20 min. After opening the left atrium, an atrial retractor was placed and the analysis and reconstruction of the mitral valve were commenced with the usual techniques. The instruments used in the left and right hand entered the thorax through the single working port, and slightly different instrument lengths were used to reduce hand collisions outside of the thorax.

All procedures were commenced by experienced surgeons.

Immediately postoperatively, the ease of the surgical procedure was recorded for educational purposes. As soon as at least one part of the operation had to be performed one-handed because of collision, friction or less freedom of the instruments, the procedure was qualified as demanding.

### Statistics

The data were analyzed using SPSS 27 System software (SPSS Inc., Chicago, IL, USA). *p*-values below 0.05 were considered to indicate statistical significance. All values were expressed as mean ± standard error of mean.

The data between the groups were compared using the student’s *t*-test for continuous variables.

## 3. Results

Out of the 79 patients, 64 patients were scheduled for TE-MVR and 15 patients were deemed to be unsuitable for TE-MVR on clinical and experience grounds because of overly narrow space (*n* = 7) and/or anticipated complex repair techniques (complex repair in Barlow disease, *n* = 2; calcified subvalvular apparatus or annulus, *n* = 2; unsuitable peripheral access due to soft plaques and overly narrow groin vessels, *n* = 1; fragile tissue, *n* = 1). They instead underwent repair through sternotomy (MVR-open).

The repair rate for all patients was 98.7%. In one patient, the repair was unsuccessful, and the mitral valve had to be replaced by a biological valve. In the TE-MVR group, the conversion rate to sternotomy was 3.1%, which affected one patient due to a further repair attempt and another patient due to severe pleural adhesions that impeded the lung’s collapse. The 30-day survival was 100%.

The procedural data of both groups are given in Table 2.

In a retrospective analysis, the DSS was 130.4 ± 18.8 mm and 108.1 ± 17.3 mm, the DSM 70.7 ± 12.1 mm and 58.5 ± 13.6 mm and the DSA 31.6 ± 8.5 and 22.3 ± 9.8 mm in the TE-MVR and MVR-open, respectively (*p* < 0.001). All the other parameters were not statistically significantly different between the two groups (Table 3).

Out of the 63 patients who underwent TE-MVR, 21 patients were found to be technically demanding due to friction and less freedom to move the instruments, necessitating one-handed maneuvers during some parts of the surgery. These were classified as TE-MVR-demanding, whereas the remaining patients were classified as TE-MVR-easy. In a retrospective analysis, none of the measured morphometric parameters showed a significant difference between the TE-MVR-demanding and TE-MVR-easy patients; thus, they were not able to predict an arduous operation.

Since the morphometric parameters influence each other, a composite parameter was calculated: DSS plus 4xICS minus DNM. It was 53.3, 39.8 and 25.6 for TE-TMReasy, TE-TMRdemanding and MVR-open, respectively (*p* < 0.05 and *p* < 0.01) (Table 3).

## 4. Discussion

The totally endoscopic approach is on the rise to become the new standard in mitral valve surgery [8,9]. Its techniques have been refined over the last decade [6,10]. The implementation of 3D endoscopy is a major asset in the surgical armory, allowing for totally endoscopic procedures with single working port access [5,6,7]. Whereas its visualization is far better than in even mitral valve procedures through sternotomy, chest size and narrow spaces may hamper the freedom of instrument movement a lot.

From surgical routine, it is well known that two different morphological entities can be challenging for a totally endoscopic approach. On the one hand, in shallow thoraces, the limited space within can be very difficult, sometimes virtually impossible, to operate on because the freedom of movement is constricted. Additionally, exposing the mitral valve with an atrial retractor can squeeze the ascending aorta to the sternum. This can lead to a distortion of the antegrade cardioplegia line or aortic valve insufficiency. In both cases, the release of the atrial retractor might be necessary during the administration of antegrade cardioplegia. Although tall, skinny patients with a low BMI are more likely to present with narrow space in their thoracic cavity, physiognomy and appearance can be misleading. On the other hand, barrel-like thoraces provide plenty of space but can be cumbersome due to long distances to the mitral valve. This is why we believe that a preoperative CT scan must be mandatory for preoperative screening, besides the established reasons such as the detection of calcifications, hard and soft plaques and vascular access.

However, the reason on clinical grounds why patients were deemed to be unsuitable for a totally endoscopic approach was overly narrow space in conjunction with anticipated complex repair techniques in every case. In this respect, DSS and DSM, measured at the level of the mitral valve, are the parameters that best reflect this anatomical aspect.

A third aspect is the working port since it comprises the pivotal point of the instrument’s movements. In totally endoscopic surgeries with three- to four-centimeter skin incisions without rib spreading, the width of the intercostal space is an important factor.

The angle at which the plane of the mitral valve can be approached, as well as the rotatory movement capacity at the tip of the instruments, are further issues in minimally invasive surgery with limited access. Besides the aforementioned parameters, we also measured from an axial view the angle between the mitral valve plane and the horizontal (AMH). Although the left atrium was lifted up with a left atrial retractor, an acute angle might negatively influence the accessibility of the mitral valve. In addition, we measured the angle between the mitral plane and the line from the working port to the mitral valve (ANM) to perceive any influence upon the surgical sequence. Although we anticipated more difficulty in the mitral valve exposition and approach, this did not hold true. The left atrium could be lifted up considerably, thus ameliorating an unfavorable angle of the mitral plane. In shallow thoraces, however, the length of the atrial retractor blade had to be shorter in tendency, since its heel could interfere with the instruments. In contrast, in the retrospective analysis, none of these parameters showed any significant influence on the procedural data, not even on soft, subjective parameters, such as the ease of the procedure.

The simple distances from the sternum to the spine (DSS) and from the sternum to the mitral anulus (DSM) seem to be the most helpful parameters in predicting whether a totally endoscopic approach is feasible with a low risk of conversion. Our conversion rate was low with two patients only. In one of these, the reason for the conversion was the impossibility of collapsing the lungs. That is why the conversion of only one patient (1.5%) could be attributed on anatomical grounds. Bal et al. described a similar finding, where the distance between the sternum and the spine was predictive of conversion to sternotomy [11].

Nevertheless, it has to be considered that the term minimally invasive cardiac surgery is still not well defined in our community. A wide variety of reported procedures exist, ranging from totally endoscopic robotic procedures [10] through four 8 mm ports to minithoracotomy with rib spreading and direct vision [12] and all shades in between [6,8,13,14,15]. The size and the width of the thoracotomy and the number of additional stab incisions are important factors, grossly influencing the ease of the procedure. The more entry space is created, the less important are the size and depth of the thorax or other morphological aspects, and preoperative planning plays a minor role. Hence, the results of our study are limited to totally endoscopic mitral valve repair with a 3 cm single working port, a 10 mm port for the 3-D camera, a stab incision for the Chitwood clamp and a stab incision for the atrial retractor, thus being probably more dependent on space issues and procedural planning. In all our procedures, sizing the mitral valve annulus, leaflet length and trigon-to-trigon distance with the commercial sizers of the companies was impossible because they simply did not fit through the single working port. This is why we took all our valve measures with threads, hand-cut paper rulers and hand-cut paper replicas routinely, which fit through the ports nicely.

Since the morphometric parameters influenced each other, a composite parameter was calculated, consisting of the available space in the thorax (DSS), the available space of the working port (ICS) and the distance from the working port to the mitral valve (DNM). From clinical experience, we estimated that the width of the intercostal space was about one half to two thirds as important as the depth of the thorax (DSS). Presuming this loading, we had to correct the absolute number of the ICS. Typically, the DSS ranged from 83 to 187 mm and the ICR from 7 to 26 mm, and this is why we multiplicated the ICS by four to correct for the quantity difference. This resulted in the following formula: DSS plus 4xICS minus DNM. In the retrospective analysis, this composite parameter was found to be significantly different between TE-MVR and MVR-open. In addition, it was significantly different between the TE-MVR-demanding and TE-MVR-easy groups.

In general, a DSS of less than 10 cm can be considered highly demanding and is likely to be associated with prolonged cardiopulmonary bypass and aortic cross-clamp times, potentially leading to inferior outcomes, particularly in complex mitral valve pathologies. It is important to emphasize that mitral valve repair should always remain the primary objective and should not be compromised in favor of a smaller incision or a minimally invasive approach. Therefore, we do not recommend a fully endoscopic approach for patients with a DSS < 10 cm, particularly for less experienced surgeons. Thus, it may help to select proper cases for less experienced or junior surgeons at the beginning of their career.

Since all our patients routinely undergo CT scans to assess cannulation sites (including femoral artery diameter and calcification), aortic calcification for cross-clamping evaluation and coronary artery calcification (coronary CT scan), this manuscript primarily focuses on CT-based parameters and measurements. However, non-radiological diagnostic tools, such as 3D-Laser Scanner, white light scanning, transesophageal echocardiography, etc. [16,17,18,19], may also become valuable in the field of cardiac surgery with increasing experience and adaptation.

It can be discussed whether the upfront decision to perform a sternotomy in patients who were deemed to be unsuitable for a totally endoscopic approach on clinical grounds is justifiable. We chose to do so to come up with a clear distinction between these two morphological entities. As a result of our findings, however, we changed our strategy: In patients with shallow thoraces and narrow intercostal spaces, we stayed with a minimally invasive approach, but performed a skin incision one to two centimeters longer and moved the working port, depending on their most likely anatomy, more laterally to achieve better angulations with the instruments. Still, no rib spreader was used, no direct vision was applied and the operation was commenced by 3D endoscopic vision only. Certainly, we no longer consider this approach a totally endoscopic procedure since the working port is significantly larger, allowing ring sizers to fit through it. We term it an endoscopic approach, still without rib spreading and no possibility of direct vision.

We acknowledge this study’s limitations, particularly the significant impact of preoperative decision-making. However, a key strength is that despite being based on experience, the established relative values or thresholds could help guide future research and contribute to the standardization of surgical training.

## 5. Conclusions

In conclusion, individual surgical skills and a long history of expertise are mandatory to achieve excellent results with a low conversion and high repair rate for totally endoscopic mitral valve repair.

DSS is an easy measure and showed excellent predictability if a totally endoscopic non-robotic approach was possible with a low conversion rate.

For educational purposes, the composite morphometric parameter (DSS + 4xICS-DNM) may be a more valid, easy and strong parameter. It allows the prediction of the possibility of TE-MVR with a low conversion rate and high success rate, as well as of the ease of TE-MVR, which is of utmost importance in a training program.

## Figures and Tables

**Figure 1 jcm-14-02581-f001:**
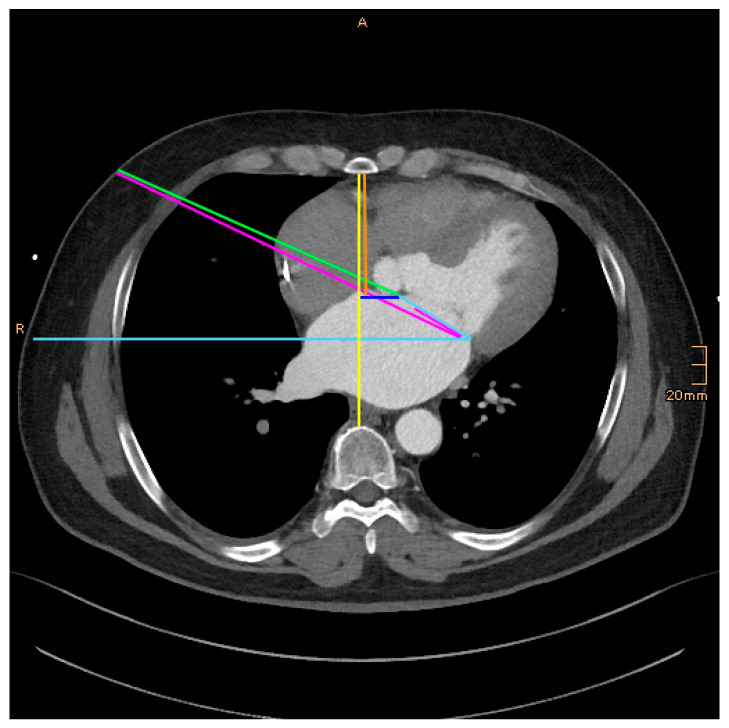
Measurements. DSS–Distance between the sternum and the spine: yellow. DSM–Distance between the sternum and the anterior annulus (A2) of the mitral valve: orange. DNM–Distance between the nipple and the anterior mitral anulus: green. DVM–Distance to the left between the vertical and the anterior mitral anulus (DVM): dark blue. AMH–Angle between the mitral valve and the horizontal: light blue. ANM–Angle between the mitral plane and the line from the working port to the mitral valve: magenta.

**Figure 2 jcm-14-02581-f002:**
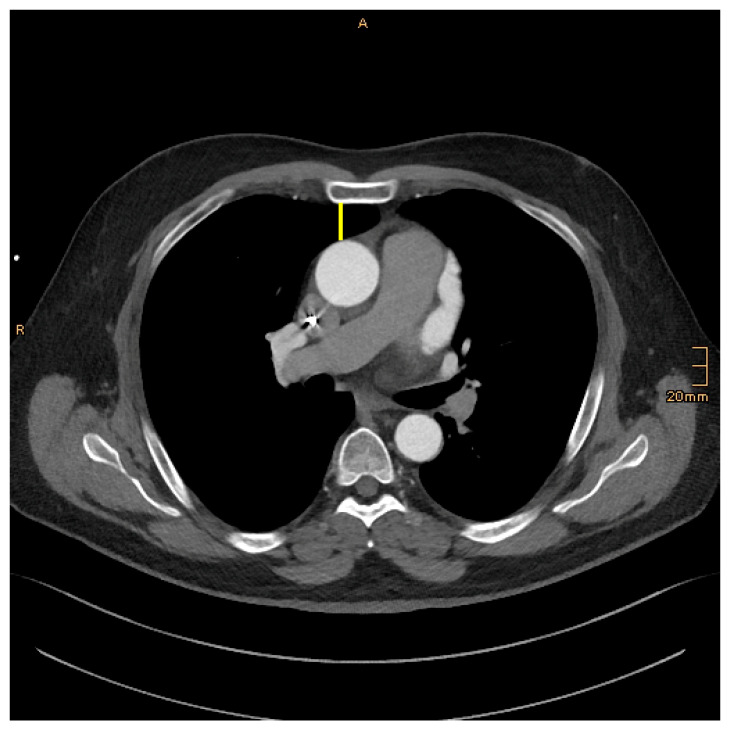
DSA–Distance between the sternum and aorta: yellow.

**Figure 3 jcm-14-02581-f003:**
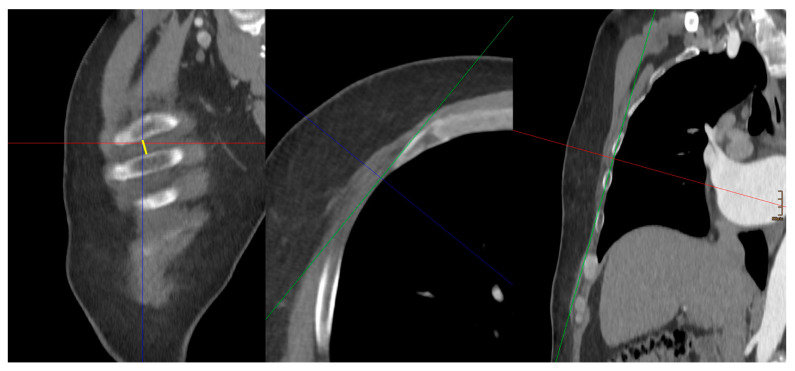
ICS–Width of the intercostal space: yellow. The red, green, and blue lines represent the orthogonal planes generated by the Multi-Planar Re-construction (MPR) plugin, facilitating three-dimensional assessment of the CT dataset.

**Table 1 jcm-14-02581-t001:** Patient characteristics.

	Mean ± stddev	Range
Gender (male/female)	51/28	
Lvef < 50% (*n*)	10	
Papsys 31–55 mmhg (*n*)	33	
Papsys > 55 mmhg (*n*)	6	
Chronic lung disease (*n*)	10	
Age (years)	59.6 ± 12.3	27.9–82.2
Height (m)	1.75 ± 0.1	1.48–1.90
Weight (kg)	76.3 ± 14.5	46–122
BMI	24.8 ± 3.7	19–34
Es log	3.0 ± 2.2	1.5–14
Es ii	1.4 ± 0.9	0.5–4.3
Creatinine (mg/dL)	0.88 ± 0.17	0.65–1.29
Creatinine clearance mL/min	94.4 ± 28.6	36–178

**Table 2 jcm-14-02581-t002:** Procedural data.

	TE-MVR	MVR-Open
MVR	63 (98.4%)	15 (100%)
annuloplasty	63 (98.4%)	15 (100%)
loops	47 (73.4%)	11 (73.3%)
loops number	4.3 ± 1.3	4.1 ± 0.9
triangular resection	1 (1.6%)	3 (20%)
cleft closure	9 (14.1%)	0
chordae transfer	2 (3.2%)	1 (6.7%)
ring decalcification	0	2 (13.4%)
TVR	6 (9%)	8 (53%)
ASD	9 (14%)	1 (6%)
MAZE	15 (13%)	6 (40%)

**Table 3 jcm-14-02581-t003:** Morphometric parameters between the groups.

	TE-MVR	MVR-Open	*p*
DSS (mm)	130.4 ± 18.8	108.1 ± 17.3	0.00007
DSM (mm)	70.7 ± 12.1	58.5 ± 13.6	0.0010
DSA (mm)	31.6 ± 8.5	22.3 ± 9.8	0.0004
DNM (mm)	147.2 ± 13.5	147.5 ± 15.4	0.9413
DVM (mm)	14.2 ± 7.7	16.4 ± 8.8	0.3284
AMH (degree)	35.6 ± 6.5	37.7 ± 6.2	0.2469
ICS (mm)	16.5 ± 3.7	16.2 ± 4.6	0.8014

## Data Availability

The data presented in this study are available on request from the corresponding author due to specific data policy of the institution.

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
