# Peer review of "Morphometric Measurements Prior to Totally Endoscopic Mitral Valve Repair: Technical and Educational Aspects"

_jcm, 2025, doi:10.3390/jcm14082581_

Round 1

Reviewer 1 Report

Comments and Suggestions for Authors

Congratulations to the authors on their work.

I have two comments that I believe will enhance the quality and clarity of this study:

  1. In the results section, you state that patients undergoing MVR-open were selected based on experience due to limited space or a complex procedure. However, the study itself also measures and links similar factors, including the anteroposterior diameter and "the demanding repair" as an outcome, which are inherently confounded at the start.

  2. To maximize the study’s impact, I suggest:

    • Proposing a relative cutoff / or interval for clinical decision making, especially in terms of the suggested morphometric parameter. Other patient-specific characteristics, procedural complexity and predicted successful repair, and surgical expertise are still important. This could be further validated in future studies. 
    • Adding a limitation paragraph in your discussion. Acknowledging the study’s limitations, particularly the strong influence of preoperative decision-making. The strength is that, even if it is experience-based, defining relative values or thresholds could guide future research and standardization efforts for future surgical training. 

Author Response

We sincerely appreciate your valuable feedback and constructive suggestions, which have significantly contributed to improving our manuscript. Below, we address your comments in detail:

Comment 1: Confounding Factors in Patient Selection and Outcome Measurement

“In the results section, you state that patients undergoing MVR-open were selected based on experience due to limited space or a complex procedure. However, the study itself also measures and links similar factors, including the anteroposterior diameter and ‘the demanding repair’ as an outcome, which are inherently confounded at the start.”

You highlighted the potential confounding between patient selection criteria and study outcome measurements, particularly regarding the anteroposterior diameter and the complexity of mitral valve repair.

Response:
We acknowledge this important point and have revised the discussion to clarify the distinction between selection criteria and the study’s morphometric findings. Additionally, we now propose a relative cutoff range for clinical decision-making based on our morphometric data. While patient-specific factors, procedural complexity, and surgical expertise remain crucial determinants, these measurements could provide an additional objective parameter to assist in preoperative planning. We have also emphasized the need for future studies to further validate these thresholds in diverse patient populations and surgical settings.

Thus, we have added the following paragraph:
"In general, a DSS of less than 10 cm can be considered highly demanding and is likely to be associated with prolonged cardiopulmonary bypass and aortic cross-clamp times, potentially leading to inferior outcomes, particularly in complex mitral valve pathologies. It is important to emphasize that mitral valve repair should always remain the primary objective and should not be compromised in favor of a smaller incision or minimally invasive approach. Therefore, we do not recommend a fully endoscopic approach for patients with a DSS < 10 cm, particularly for less experienced surgeons."

Comment 2: Adding a Limitations Paragraph

You suggested explicitly acknowledging the strong influence of preoperative decision-making as a limitation and emphasizing the study’s strengths in defining experience-based relative values.

Response:
We have now included a dedicated limitations paragraph in the discussion:
"We acknowledge the study’s limitations, particularly the significant impact of preoperative decision-making. However, a key strength is that, despite being based on experience, establishing relative values or thresholds could help guide future research and contribute to the standardization of surgical training."

We appreciate your insightful comments, which have strengthened the clarity and applicability of our study. Please let us know if any further modifications are needed.

Reviewer 2 Report

Comments and Suggestions for Authors

In this interesting paper, the authors analyzed several morphometric measurement tools for evaluating the feasibility of totally endoscopic mitral valve repair (TE-MVR).

The narrower the distance between the sternum and the spine (the conventional Haller index) measured on CT scan, the higher the risk of conversion to sternotomy (MVR-open).

In the authors' findings, the distances between the sternum and the spine and between the sternum and the mitral annulus were the most helpful parameters in predicting the feasibility of TE-MVR. The authors also tested an innovative composite morphometric parameter, to predict the TE-MVR feasibility.

The manuscript is well written and very interesting for clinical cardiologists.

The tables and figures are clear, the references are appropriate and the conclusions correctly summarize the main findings of the study.

I have one suggestion for the authors.

In the Discussion section, on line 250, the authors could also discuss the potential usefulness of nonradiological indices that may provide an accurate assessment of the distance between the sternum and the spine (Haller index) without the CT-related ionizing exposure. Recently, some authors have validated nonradiological techniques to assess the antero-posterior (A-P) thoracic diameter in individuals with pectus excavatum, by using: 1) a rigid ruler coupled to a level (PMID: 17952321); 2) transthoracic echocardiography (PMID: 29766334); 3) three-dimensional laser scanner (PMID: 28094014), or white light scanning (PMID: 30732932). The use of these innovative nonradiological techniques should be considered for implementation in the evaluation of potential candidates to TE-MVR. Probably, individuals with a too narrow A-P thoracic diameter (<11 cm) should be excluded from the TE-MVR.

Author Response

We sincerely appreciate your positive feedback and thoughtful suggestions, which help enhance the quality and clinical relevance of our manuscript. Below, we address your comment in detail:

Comment: Discussion of Nonradiological Indices for Assessing the Anteroposterior Thoracic Diameter

You suggested expanding the Discussion section to include the potential role of nonradiological methods in assessing the anteroposterior thoracic diameter, thereby reducing reliance on CT-related ionizing radiation. Specifically, you referenced techniques such as a rigid ruler coupled with a level, transthoracic echocardiography, three-dimensional laser scanning, and white light scanning.

Response:
We appreciate this valuable insight and have now incorporated a discussion on nonradiological alternatives for evaluating the anteroposterior thoracic diameter. We acknowledge that while CT scanning remains the gold standard for precise morphometric assessment, nonradiological techniques have demonstrated promising accuracy in other clinical contexts, such as pectus excavatum assessment. Implementing these methods in preoperative screening for totally endoscopic mitral valve repair (TE-MVR) could potentially reduce radiation exposure and streamline patient selection. However, further validation is required to determine their reliability in predicting surgical feasibility.

Thus, we have added the following paragraph:
"Since all our patients routinely undergo CT scans to assess cannulation sites (including femoral artery diameter and calcification), aortic calcification for cross-clamping evaluation, and coronary artery calcification (coronary CT scan), this manuscript primarily focuses on CT-based parameters and measurements. However, non-radiological diagnostic tools may also become valuable in the field of cardiac surgery with increasing experience and adaptation."

Thank you once again for your insightful suggestion, which has strengthened the discussion of alternative assessment methods in our manuscript. Please let us know if any additional modifications are needed.

Round 2

Reviewer 1 Report

Comments and Suggestions for Authors

No further comments. 
Thanks for addressing the raised comments.